# Elevation of Anticancer Drug Toxicity by Caffeine in Spheroid Model of Human Lung Adenocarcinoma A549 Cells Mediated by Reduction in Claudin-2 and Nrf2 Expression

**DOI:** 10.3390/ijms232415447

**Published:** 2022-12-07

**Authors:** Hiroaki Eguchi, Riho Kimura, Saki Onuma, Ayaka Ito, Yaqing Yu, Yuta Yoshino, Toshiyuki Matsunaga, Satoshi Endo, Akira Ikari

**Affiliations:** 1Laboratory of Biochemistry, Department of Biopharmaceutical Sciences, Gifu Pharmaceutical University, Gifu 501-1196, Japan; 2Laboratory of Bioinformatics, Gifu Pharmaceutical University, Gifu 502-8585, Japan

**Keywords:** lung adenocarcinoma, caffeine, chemoresistance, claudin-2

## Abstract

Claudin-2 (CLDN2), a component of tight junctions, is abnormally expressed in human lung adenocarcinoma tissue. CLDN2 contributes to chemoresistance in human lung adenocarcinoma-derived A549 cells, and it may be a target for cancer therapy. Here, we found that coffee ingredients, namely caffeine and theobromine, decreased the protein level of CLDN2 in human lung adenocarcinoma-derived A549 cells. In contrast, other components, such as theophylline and chlorogenic acid, had no effect. These results indicate that the 7-methyl group in methylxanthines may play a key role in the reduction in CLDN2 expression. The caffeine-induced reduction in the CLDN2 protein was inhibited by chloroquine, a lysosome inhibitor. In a protein-stability assay using cycloheximide, CLDN2 protein levels decreased faster in caffeine-treated cells than in vehicle-treated cells. These results suggest that caffeine accelerates the lysosomal degradation of CLDN2. The accumulation and cytotoxicity of doxorubicin were dose-dependently increased, which was exaggerated by caffeine but not by theophylline in spheroids. Caffeine decreased nuclear factor-erythroid 2-related factor 2 (Nrf2) levels without affecting hypoxia-inducible factor-1α levels. Furthermore, caffeine decreased the expression of Nrf2-targeted genes. The effects of caffeine on CLDN2 expression and anticancer-drug-induced toxicity were also observed in lung adenocarcinoma RERF-LC-MS cells. We suggest that caffeine enhances doxorubicin-induced toxicity in A549 spheroids mediated by the reduction in CLDN2 and Nrf2 expression.

## 1. Introduction

Despite significant advances in treatment methods, cancer is often a fatal disease. Cancer cells are born through the gradual acquisition of genetic alterations in an organism from approximately 2000 to over 3000 every day, which are usually eliminated by immune cells [1]. However, some of these cells evade immune surveillance, leading to their survival and uncontrollable growth. Newly born cancer cells may be sensitive to anticancer drugs and radiotherapy, but they develop treatment resistance. A microenvironment is formed by cancer cells, endothelial cells, fibroblasts, and so on in the body, which facilitate tumor aggressiveness and chemoresistance [2].

Epithelial cells form tight junctions (TJs) at the most apical region of the lateral membranes of two adjacent cells [3]. TJs are composed of various combinations of claudins and occludin. So far, at least 27 claudin (CLDN) subtypes have been identified in mammalian cells [4,5,6]. These subtypes are specifically and selectively expressed in each tissue and may confer specific properties. Abnormal expressions of CLDN subtypes have been reported in various solid tumor tissues. CLDN2 is highly expressed in human solid tumors, including lung adenocarcinoma [7], colon [8], liver [9], and stomach cancers [10]. CLDN2 forms a pore for cations [11] and is a barrier against small-sized hydrophilic substances [12]. In addition, CLDN2 affects the proliferation of cancer cells. Cell proliferation is suppressed by the silencing of CLDN2 expression in human lung adenocarcinoma A549 [13] and colon cancer Caco-2 cells [14].

We reported that CLDN2 enhances chemoresistance in A549 spheroid cells, an in vitro model of the microenvironment [15]. Stress generation mechanisms in the microenvironment are not fully understood. In the microenvironment, the hypoxia-inducible factor-1α (HIF-1α) and nuclear factor-erythroid 2-related factor 2 (Nrf2) pathways can be activated in response to hypoxic conditions and the generation of reactive oxygen species (ROS) [16]. These stress factors may have a key role in the malignant transformation of cancer cells. Therefore, it is important to develop food ingredients and drugs that can inhibit stress response and/or decrease CLDN2 expression in lung adenocarcinoma cells. The expression of CLDN2 in A549 cells is controlled at the transcriptional, post-transcriptional, and post-translational steps. So far, we have reported that some flavonoids can reduce CLDN2 expression in A549 cells. Quercetin decreases CLDN2 expression mediated by an upregulation of miR-16 expression and the instability of claudin-2 mRNA [17]. In addition, intracellular signaling pathways, including the mitogen-activated protein kinase (MEK)/extracellular-signal-regulated kinase (ERK), phosphoinositide 3-kinase (PI3K)/Akt, and STAT3 pathways, are involved in the transcriptional regulation of CLDN2 [7,18,19]. Thus, the mechanisms of flavonoids in CLDN2 expression have been well characterized in our previous study. However, the effects of other food ingredients used for human consumption have not been sufficiently investigated.

Coffee is one of the most popular and widely consumed beverages around the world. It contains various chemical compounds, such as caffeine, chlorogenic acid, kahweol, and cafestol [20]. A meta-analysis of prospective studies shows a protective effect of coffee consumption on colorectal cancer in the USA [21]. Caffeine exerts a broad range of effects, including antagonizing adenosine G-protein-coupled receptors, inhibiting phosphodiesterase, and sensitizing calcium channels [22]. Caffeine enhances anti-tumor immune responses mediated through a reduction in PD1 expression in T lymphocytes [23]. Caffeic acid and chlorogenic acid have been reported to have anticancer effects mediated by antioxidative abilities [24]. However, another anticancer mechanism of coffee ingredients is not well understood. In the present study, we found that caffeine and theobromine can reduce CLDN2 expression in A549 cells. Therefore, the effects of these coffee ingredients on mRNA and protein expression levels of CLDN2 were determined using real-time polymerase chain reaction (PCR) and Western blotting analyses, respectively. The paracellular barrier function of CLDN2 was evaluated using transepithelial electrical resistance (TER) and fluxes of small molecular compounds. The sensitivity against anticancer drugs was investigated using a spheroid model.

## 2. Results

### 2.1. Decrease in the Protein Level of CLDN2 in A549 Cells by Caffeine and Theobromine

Caffeine, theophylline, and theobromine are known as methylxanthines that are naturally present in foods, including coffee [22]. A549 cells cultured on flat-bottomed plates were treated with coffee ingredients for 24 h. The tested methylxanthines did not show cytotoxicity until a concentration of 200 µM in our experimental conditions (Figure 1A). The protein level of CLDN2 was dose-dependently decreased by caffeine and theobromine, not by theophylline (Figure 1B). In contrast, the protein level of CLDN1 was not affected by methylxanthines. Chlorogenic acid, kahweol, and cafestol are also contained in coffee extracts. Chlorogenic acid at a concentration of below 200 µM did not induce cytotoxicity, nor did kahweol and cafestol at a concentration of below 10 µM (Figure 2A). The protein levels of neither CLDN1 nor CLDN2 were changed by these compounds (Figure 2B).

### 2.2. Effects of Coffee Ingredients on the Cellular Localization of CLDN2

So far, we reported that CLDN2 is mainly colocalized with zonula occludens-1 (ZO-1) at the cell–cell border area in A549 cells using fluorescence measurements [7]. The red signal of CLDN2 expression was detected at the cell–cell border area, which was diminished by caffeine and theobromine (Figure 3). On the other hand, the green signal of ZO-1 expression was unchanged. Additionally, other coffee ingredients had no effect on the cellular localization of CLDN2 and ZO-1. The effects of caffeine and theobromine on CLDN2 expression found in immunofluorescence assays coincide with those by Western blotting.

### 2.3. Decrease in the Protein Stability of CLDN2 by Caffeine

The mRNA level of CLDN2 was decreased by both caffeine and theobromine, but not by theophylline (Figure 4). The mRNA level of CLDN1 was unaffected by these methylxanthines. As shown in Figure 1B, the protein level of CLDN2 was significantly reduced by 200 µM caffeine (approximately 80%), whereas it had less effect on mRNA levels (approximately 20%). Therefore, we examined the effects of caffeine on the protein stability of CLDN2 using treatment with cycloheximide (CHX), a translational inhibitor. The decrease in the protein levels of CLDN2 was enhanced by caffeine (Figure 5A). However, the caffeine-induced CLDN2 reduction was rescued by chloroquine, a lysosome inhibitor, but not by lactacystin, a proteasome inhibitor (Figure 5B). In immunofluorescence measurements, CLDN2 was colocalized with Lamp-1, a lysosome marker, in caffeine- and chloroquine-treated cells (Figure 5C). The stability of CLDN2 protein in TJs is upregulated by the phosphorylation of Ser208 [25]. The phosphoserine level of CLDN2 was significantly decreased by caffeine (Figure 5D).

### 2.4. Effect of Caffeine on Proliferation, Migration, and Paracellular Barrier Function

So far, we reported that CLDN2 is involved in the regulation of cell proliferation and migration in A549 cells [13,15]. Cell numbers were decreased in caffeine-treated cells compared with vehicle-treated cells (Figure 6A). In a wound-healing assay, the recovery rate of the wound area was decreased by treatment with caffeine (Figure 6B). These results indicate that caffeine may suppress cell proliferation and migration. TJs regulate paracellular permeability across epithelia, and the expression of CLDN2 is involved in the regulation of paracellular permeability to Na^+^ and water [11]. We investigated the function of the TJ barrier by measuring the TER and transepithelial flux of small molecules, including doxorubicin (DXR) and lucifer yellow (LY). TER was significantly increased by caffeine, whereas theophylline had no effect (Figure 6C). The transepithelial permeabilities of both DXR and LY were enhanced by caffeine but not by theophylline (Figure 6D). The data for caffeine were consistent with previous reports on CLDN2 knockdown experiments [15].

### 2.5. Inhibition of Nrf2-Mediated Stress Signaling by Caffeine in Spheroids

Nutrient gradients, hypoxia, and oxidative stress must be involved in the acquisition of chemoresistance in the tumor microenvironment [16]. The 3D spheroid model mimics the tumor microenvironment, which is useful to the investigation of the mechanisms of chemoresistance. Caffeine significantly decreased the fluorescence intensities of hypoxia probe LOX-1 and ROS probe CellRox Deep Red (Figure 7A,B), whereas theophylline had no effect. Caffeine significantly reduced the protein level of Nrf2, whereas it had no effect on that of HIF-1α (Figure 7C). The mRNA levels of Nrf2 target genes, such as HO-1, NQO-1, and GCLM, were significantly decreased by caffeine (Figure 7D). Our data indicate that caffeine may suppress the Nrf2-dependent stress signal mediated by a reduction in CLDN2 expression.

### 2.6. Enhancement of Anticancer Drug-Induced Toxicity by Caffeine

Spheroids were treated with the indicated concentrations of DXR for 1 h, followed by a measurement of the fluorescence intensity of DXR. DXR accumulation in spheroids was dose-dependently increased, which was significantly enhanced by caffeine (Figure 8A). In contrast, theophylline had no effect. The viability of spheroid cells was dose-dependently decreased by DXR and cisplatin (CDDP), which was exaggerated by caffeine (Figure 8B,C). Our data indicate that caffeine may enhance toxicity against anticancer drugs mediated by the reduction in CLDN2 expression in A549 spheroid cells. Next, we performed similar studies using another adenocarcinoma cell line, RERF-LC-MS cells. Caffeine decreased the protein level of CLDN2 in a dose-dependent manner without affecting CLDN1 expression (Figure 9A). Both the accumulation and toxicity of DXR were increased in a dose-dependent manner, and were further enhanced by caffeine (Figure 9B,C). These results are consistent with those in A549 cells.

## 3. Discussion

CLDN2 is involved in the upregulation of proliferation, migration, and chemoresistance in human lung adenocarcinoma cells [13,15]. Some flavonoids, including kaempferide, quercetin, and chrysin, can reduce CLDN2 expression in A549 cells [17,18,19], but the effects of other food ingredients have not been sufficiently investigated. Here, we found that caffeine, a component of coffee, can reduce the protein level of CLDN2 (Figure 1). The protein level of CLDN2 was also decreased by theobromine, but not by theophylline. Both caffeine and theobromine possess a methyl group at the 7th position in the xanthine structure, but theophylline does not. Our results suggest that the methylation of xanthine plays an important role in the suppression of CLDN2 expression. Caffeine also enhances the bradykinin-induced reduction in CLDN5 expression in a primary culture of rat-brain microvascular endothelial cells [26]. However, there are no reports concerning other CLDNs. This is the first report showing that caffeine can reduce CLDN2 expression in lung adenocarcinoma.

Caffeine has an antagonistic effect on adenosine by binding to adenosine receptors (Ki values: ~10 µM) and theophylline is slightly more active [27]. In contrast, theobromine is markedly less active on adenosine receptors (Ki values: >100 μM). The A2B adenosine receptor is highly expressed in A549 cells and involved in epithelial–mesenchymal transition [28]. Kitabatake et al. [29] reported that the involvement of the adenosine A2B receptor in the radiation-induced translocation of epidermal growth factor receptor and DNA damage response leads to radioresistance in human lung cancer cells. In contrast, the A2B adenosine receptor subtype is also considered an important target for cancer therapy [30]. However, the adenosine receptor is considered unlikely to be involved in the regulation of CLDN2 expression because (1) CLDN2 expression is decreased by caffeine without adenosine stimulation, (2) CLDN2 expression was not decreased by theophylline, and (3) caffeine showed no effect on CLDN2 expression at a concentration of below 50 μM.

The protein level of CLDN2 may be controlled at the transcriptional, post-transcriptional, and post-translational steps. The effect of caffeine (200 μM) on the mRNA level of CLDN2 (approximately 20%) was less than that on the protein level (approximately 80%) (Figure 1B and Figure 4). The protein-stability assay showed that caffeine significantly accelerated the degradation of the CLDN2 protein (Figure 5A). Furthermore, the caffeine-induced reduction in the CLDN2 protein was significantly inhibited by chloroquine (Figure 5B). These results suggest that caffeine enhances the degradation of the CLDN2 protein in the lysosomal proteolytic pathway.

The stability of the CLDN2 protein in TJs is upregulated by the phosphorylation of Ser208 [25]. The phosphoserine level of CLDN2 was decreased by caffeine (Figure 5C), suggesting that the stability of the CLDN2 protein in TJs is attenuated by caffeine. The phosphorylation status of CLDN2 was downregulated in a cAMP/protein kinase A (PKA)-dependent manner in Madin–Darby canine kidney cells. Dephosphorylated CLDN2 protein is translocated to the lysosome. CLDN2 is distributed into the lysosome in caffeine- and chloroquine-treated cells (Figure 5C). Caffeine increases PKA activity mediated by inhibiting phosphodiesterase, an enzyme involved in the degradation of cAMP [31]. At present, it is unknown whether caffeine affects the cAMP/PKA-dependent signaling pathway in A549 cells or not. Another explanation is the involvement of protein phosphatases. We have reported that caffeic acid phenethyl ester decreases the phosphoserine level of CLDN2, which is inhibited by cantharidin, a protein phosphatase inhibitor [32]. The activity of protein phosphatase 2A is increased by caffeine in C2C12 myotube cells [33] and human leukemia U937 cells [34]. Caffeine may decrease the stability of the CLDN2 protein mediated by the activation of protein phosphatases.

The toxicities of DXR and CDDP were exaggerated by caffeine in A549 spheroid cells, but not by theophylline (Figure 8). Caffeine has been reported to enhance the antitumor effect of CDDP in lung and hepatocellular tumor cells [35,36]. CLDN2 is highly expressed in human lung adenocarcinoma [7] and liver cancer cells [9]. These reports support the idea that a reduction in CLDN2 expression may be involved in the caffeine-induced rescue of chemoresistance. Cancer cells can acquire resistance to anticancer drugs mediated by various mechanisms, including drug target alteration, drug efflux, DNA repair, escape from apoptotic and necrotic cell death, switch in energy metabolism, and so on [37]. Oxidative stress and Nrf2 signaling are linked to increased chemoresistance [38]. Nrf2 is constantly ubiquitinated by keap1 and degraded by the ubiquitin–proteasome pathway in normal conditions [39]. Upon exposure to oxidative stress conditions, keap1 loses its ability to repress Nrf2 expression, thereby facilitating the accumulation of Nrf2. Caffeine decreased ROS levels and Nrf2 expression in A549 spheroid cells, leading to a reduction in the expression of endogenous antioxidant genes (Figure 7). Caffeine may suppress chemoresistance mediated by the reduction in ROS stress. We recently reported that CLDN2 decreases the intracellular content of glucose in A549 spheroid cells [40]. Glucose deprivation shifts energy production from glucose metabolism to mitochondrial respiration. The activation of mitochondrial respiration can enhance the production of ROS. Caffeine may alleviate chemoresistance mediated via the suppression of CLDN2-dependent mitochondrial respiration.

A high caffeine content is found in many over-the-counter products, such as energy drinks, exercise supplements, health food products, stay-awake pills, and so on. We found that caffeine (>100 μM) may suppress the chemoresistance of lung adenocarcinoma, but the use of high doses of caffeine increases the risk of cardiovascular diseases. Fatalities from caffeine intoxication have been reported with plasma caffeine concentrations of <40 mg/L [41], which is approximately equal to concentrations of 200 μM, showing the biological effect in the present study. Lethal doses of caffeine in adults have been reported at blood concentrations of 80 to 100 μg/mL, which can be reached with an intake of around 10 g or greater [42]. Therefore, we have to be careful not to overdose.

## 4. Materials and Methods

### 4.1. Materials

Caffeine, kahweol, and cafestol were purchased from Cayman Chemical Company (Ann Arbor, MI, USA). Theophylline and theobromine were obtained from Tokyo Kasei Kogyo (Tokyo Japan). Dimethyl sulfoxide was the vehicle used for the dissolution coffee ingredients. Rabbit anti-CLDN1, rabbit anti-CLDN2, and mouse anti-ZO-1 antibodies were obtained from Thermo Fisher Scientific (Rockford, IL, USA). Goat anti-β-actin antibody, DXR, and chlorogenic acid were obtained from Santa Cruz Biotechnology (Santa Cruz, CA, USA), Fujifilm Wako Pure Chemical Industries (Osaka, Japan), and LKT Laboratories (St. Paul, MN, USA). All other reagents were of the highest purity available.

### 4.2. Cell Culture

A549 and RERF-LC-MS cells derived from human lung adenocarcinoma were obtained from the RIKEN BRC through the National Bio-Resource Project of the MEXT (Ibaraki, Japan). The cells were grown in Dulbecco’s modified Eagle’s medium (Sigma-Aldrich, St. Louis, MO, USA) as described previously [43]. In a two-dimensional (2D) model, the cells were grown on flat-bottomed 96-well plates, 6 cm and 10 cm dishes. In a three-dimensional (3D) model, the cells were grown on PrimeSurface 96U plates (Sumitomo Bakelite, Tokyo, Japan). After culturing for 72 h, the cells were incubated in the absence or presence of coffee ingredients for 6 h (real-time PCR) or 24 h (Western blotting and viability assays).

### 4.3. Cytotoxicity

The viability of 2D grown cells was assessed using a Cell Counting Kit-8 (CCK-8) (Dojindo-Laboratories, Kumamoto, Japan), and the absorbance at 450 nm was measured using an iMark microplate reader (Bio-Rad Laboratories, Richmond, CA, USA). The viability of 3D grown cells was assessed using a CellTiter-Glo 3D Cell Viability Assay kit (Promega, Madison, WI, USA) and the intensity of chemiluminescence was measured using an Infinite F200Pro microplate reader (Tecan, Maennedorf, Switzerland).

### 4.4. Immunoprecipitation and Western Blotting

The preparation of cell lysates, sodium dodecyl sulfate–polyacrylamide gel electrophoresis, and Western blotting were performed as described previously [43]. In assays determining the phosphorylation level of CLDN2, the cell lysates were immunoprecipitated with anti-CLDN2 antibodies and protein G sepharose beads. The signals of the protein were normalized to the loading control (α-actin or nucleoporin p62). The optical band density was quantified using ImageJ software (National Institute of Health, Bethesda, MD, USA).

### 4.5. Immunofluorescence Measurement

Both CLDN2 and ZO-1 was immunostained with each primary antibody as described previously [18]. The cellular localization of CLDN2 and ZO-1 was analyzed using an LSM700 confocal laser microscope (Carl Zeiss, Jena, Germany).

### 4.6. RNA Isolation, Reverse Transcription, and Quantitative Real-Time PCR

Reverse transcription and quantitative real-time PCR reaction were performed using specific primers against human CLDN1, CLDN2, and β-actin as described previously [19]. The mRNA levels were compensated by β-actin.

### 4.7. Cell Proliferation and Wound-Healing Assays

After culturing for 24, 48, and 72 h, cell images were obtained using an EVOS FL Auto 2 (Thermo Fisher Scientific). Cell proliferation was estimated by counting the number of cells using CKX-CCSW software (Olympus, Tokyo, Japan). In the wound-healing assay, the cells treated with vehicle dimethyl sulfoxide or caffeine were scratched using a yellow tip. Then, cell images were obtained with a BZ-X800 fluorescence microscope (Keyence, Osaka, Japan). The wound area was calculated using ImageJ software.

### 4.8. Paracellular Permeabilities to Electrolyte Ions and Small Molecules

Cells were cultured on Transwell plates with 0.4 µm pore polyester membrane inserts (Corning Incorporated, Corning, NY, USA). The barrier function of TJs was examined by measuring TER and fluxes for LY (MW: 457) and DXR (MW: 580) as described previously [44,45].

### 4.9. Spheroid Analysis

In the 3D model, bright and fluorescence images of spheroids were obtained using a BZ-X810 fluorescence microscope (Keyence, Osaka, Japan). Hypoxia and ROS stresses were analyzed using a hypoxia probe, LOX-1, and an ROS probe, CellROX Deep Red. These fluorescence probes were incubated for 1 h at 37 °C. DXR was incubated for 1 h (fluorescence measurement assay) or 24 h (viability assay) at 37 °C.

### 4.10. Statistical Analysis

Data are presented as means ± S.E.M. Comparisons between two groups were made using Student’s *t*-test. Differences between groups were analyzed using one-way analysis of variance, and corrections for multiple comparison were made using Tukey’s multiple comparison test. Statistical analyses were performed using KaleidaGraph version 4.5.1 software (Synergy Software, Reading, PA, USA). Significant differences were assumed at *p* < 0.05.

## 5. Conclusions

We found that coffee ingredients, namely caffeine and theobromine, can decrease the protein level of CLDN2 in human lung adenocarcinoma A549 cells. This effect mainly occurs through the acceleration of protein degradation in lysosomes. The toxicity of anticancer drugs, including DXR and CDDP, was exaggerated by caffeine in A549 spheroids. Caffeine- or theophylline-rich foods for human consumption may be useful as a precaution against the malignant transformation of lung adenocarcinoma.

## Figures and Tables

**Figure 1 ijms-23-15447-f001:**
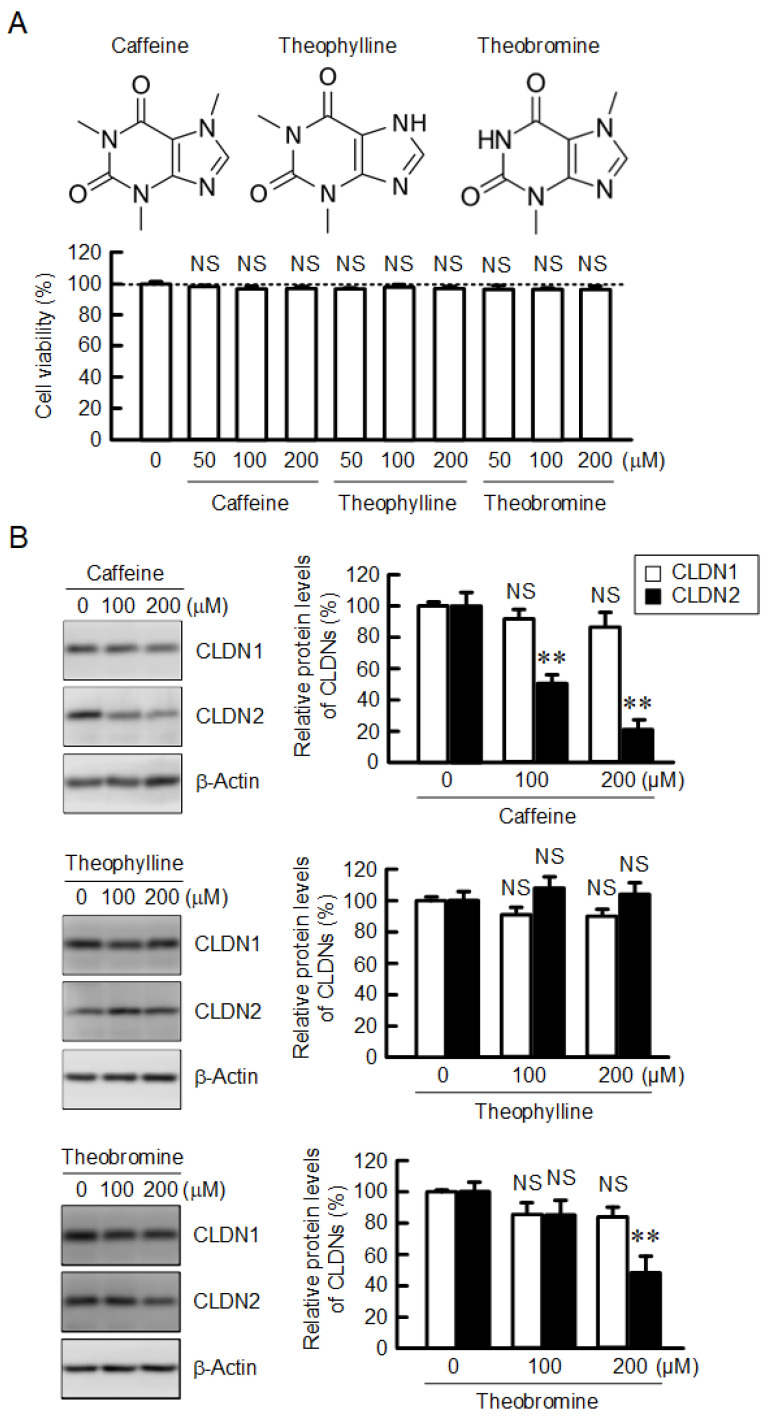
Effects of methylxanthine derivatives on cell viability and CLDN expression in A549 cells. (**A**) The structures of caffeine, theophylline, and theobromine are shown in the upper panel. Cells were incubated with caffeine, theophylline, or theobromine at the indicated concentrations for 24 h, followed by incubation with CCK-8 reagent. Cell viability is represented as a percentage relative to 0 μM. (**B**) The expression levels of CLDN1, CLDN2, and β-actin were examined by Western blotting. The protein levels of CLDN1 and CLDN2 are represented as a percentage relative to 0 μM. n = 4–7. ** *p* < 0.01 and NS *p* > 0.05 compared with 0 μM.

**Figure 2 ijms-23-15447-f002:**
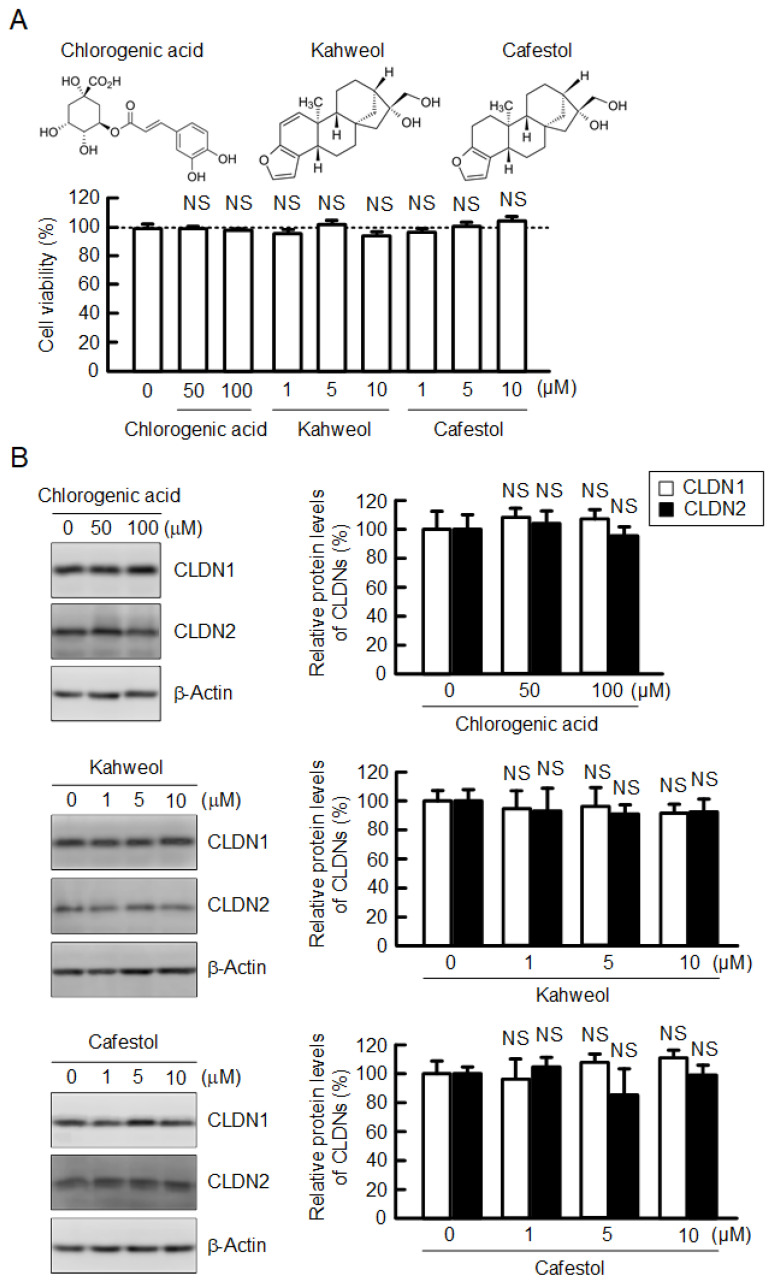
Effects of chlorogenic acid, kahweol, and cafestol on cell viability and CLDN expression. (**A**) The structures of chlorogenic acid, kahweol, and cafestol are shown in the upper panel. Cells were incubated with chlorogenic acid, kahweol, and cafestol at the indicated concentrations for 24 h, followed by incubation with CCK-8 reagent. Cell viability is represented as a percentage relative to 0 μM. (**B**) The expression levels of CLDN1, CLDN2, and β-actin were examined by Western blotting. The protein levels of CLDN1 and CLDN2 are represented as a percentage relative to 0 μM. n = 3–5. NS *p* > 0.05 compared with 0 μM.

**Figure 3 ijms-23-15447-f003:**
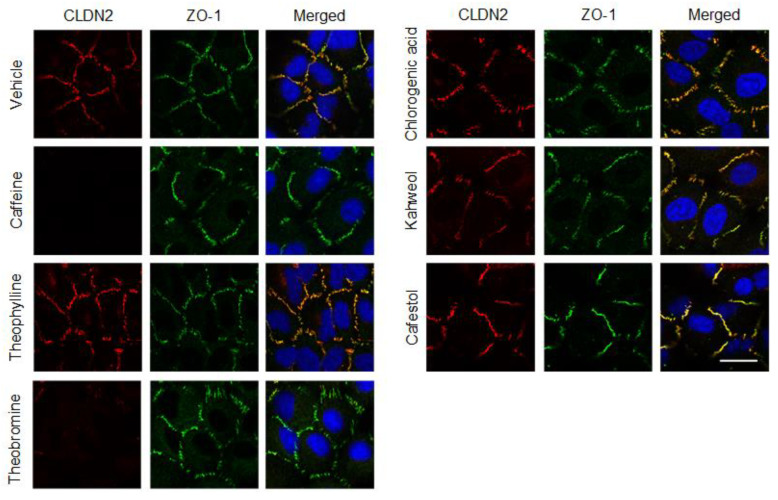
Effects of coffee ingredients on the cellular localization of CLDN2 and ZO-1. The cellular localization of CLDN2 and ZO-1 is indicated as red and green fluorescence signals, respectively. Merged images with DAPI (blue, nuclear marker) are shown on the right. The scale bar indicates 10 μm.

**Figure 4 ijms-23-15447-f004:**
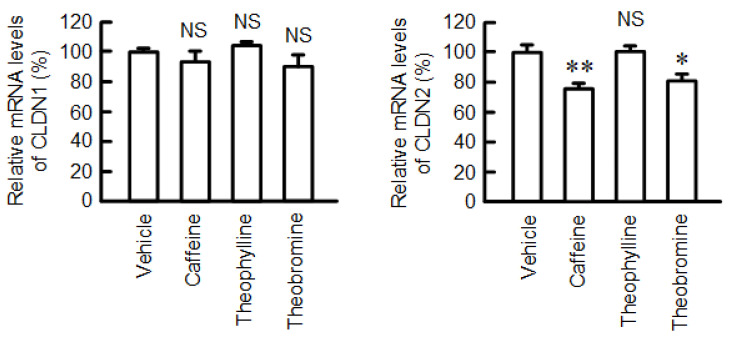
Effects of coffee ingredients on the mRNA levels of CLDNs. The mRNA levels of CLDN1 and CLDN2 were measured using real-time PCR and are represented as a percentage relative to vehicle. n = 3–4. ** *p* < 0.01, * *p* < 0.05, and NS *p* > 0.05 compared with vehicle.

**Figure 5 ijms-23-15447-f005:**
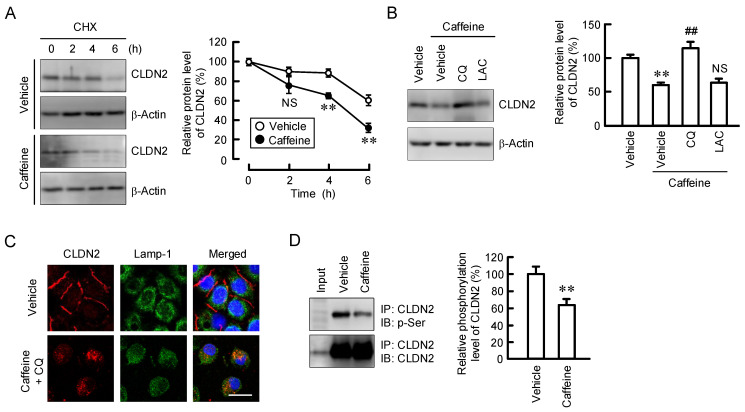
Effect of caffeine on protein stability of CLDN2. (**A**) Cells were incubated without (vehicle) or with 200 μM caffeine for the indicated periods in the presence of 3 μg/mL CHX. The protein level of CLDN2 is represented as a percentage relative to 0 h. (**B**) Cells were incubated in the absence (vehicle) and presence of 200 μM caffeine, 10 μM chloroquine (CQ), or 10 μM lactacystin (LAC) for 24 h. The protein level of CLDN2 is represented as a percentage relative to vehicle. (**C**) Cells were incubated in the absence or presence of 200 μM caffeine for 2 h. After immunoprecipitation with anti-CLDN2 antibody, aliquots were blotted with anti-phosphoserine and anti-CLDN2 antibodies. (**D**) The phosphoserine level of CLDN2 is represented as a percentage relative to vehicle. Input is loaded in the left lane. n = 5–7. ** *p* < 0.01 compared with 0 h or vehicle. ^##^
*p* < 0.01 compared with caffeine alone. NS *p* > 0.05 compared with 0 h or caffeine alone.

**Figure 6 ijms-23-15447-f006:**
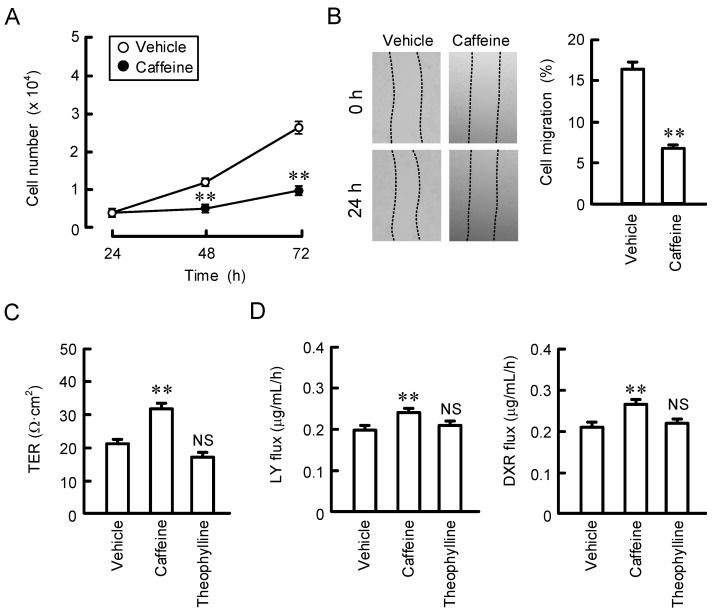
Effects of caffeine and theophylline on cell proliferation, migration, and paracellular barrier function. (**A**,**B**) Cells cultured on 6 cm dishes were incubated in the absence (vehicle) or presence of 200 μM caffeine. In the proliferation assay, cell images were obtained at 24, 48, and 72 h, and the number was calculated. In the wound-healing assay, cell images were obtained at 0 and 24 h after wound creation, and the remaining wound area was calculated. (**C**,**D**) Cells cultured on Transwell inserts were incubated in the absence (vehicle) or presence of 200 μM caffeine or theophylline for 24 h. TER was measured using a volt ohm meter. The fluxes of LY and DXR were measured using a microplate reader. Then, the solution in the basal compartment was collected, followed by a measurement of the fluorescence intensity. n = 4. ** *p* < 0.01 and NS *p* > 0.05 compared with vehicle.

**Figure 7 ijms-23-15447-f007:**
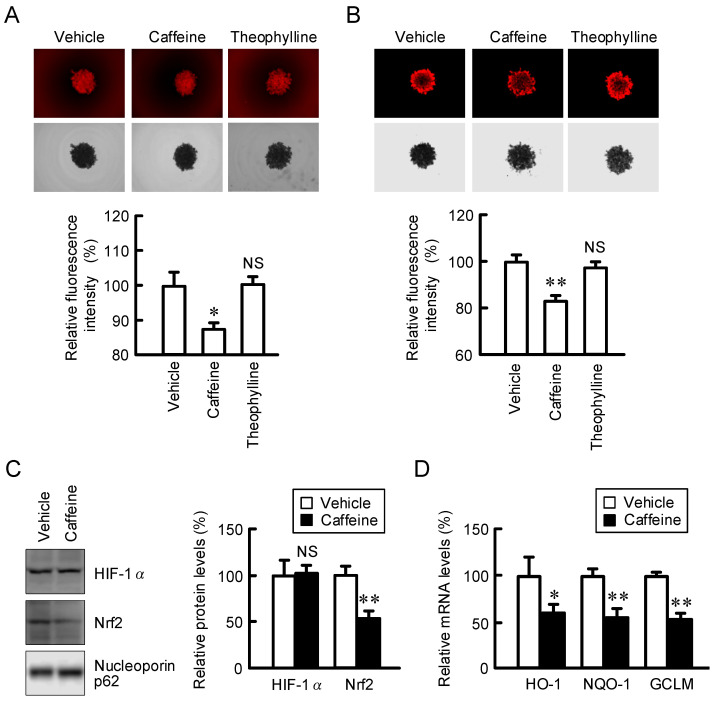
Effects of caffeine and theophylline on stress signals in spheroid cells. Spheroid cells were pre-incubated in the absence (vehicle) or presence of 200 μM caffeine or theophylline for 24 h. (**A**,**B**) The cells were incubated with hypoxia marker LOX-1 (**A**) and ROS marker CellRox Deep Red (**B**). The upper and lower panels indicate fluorescence and bright field images, respectively. The fluorescence intensity is represented as a percentage relative to vehicle. (**C**) The protein expression of HIF-1α, Nrf2, and nucleoporin p62 (a loading control) in spheroids was examined using Western blotting and represented as a percentage relative to vehicle. (**D**) The mRNA expression levels of HO-1, NQO-1, and GCLM in spheroids were examined using real-time PCR and represented as a percentage relative to vehicle. n = 3–7. ** *p* < 0.01, * *p* < 0.05, and NS *p* > 0.05 compared with vehicle.

**Figure 8 ijms-23-15447-f008:**
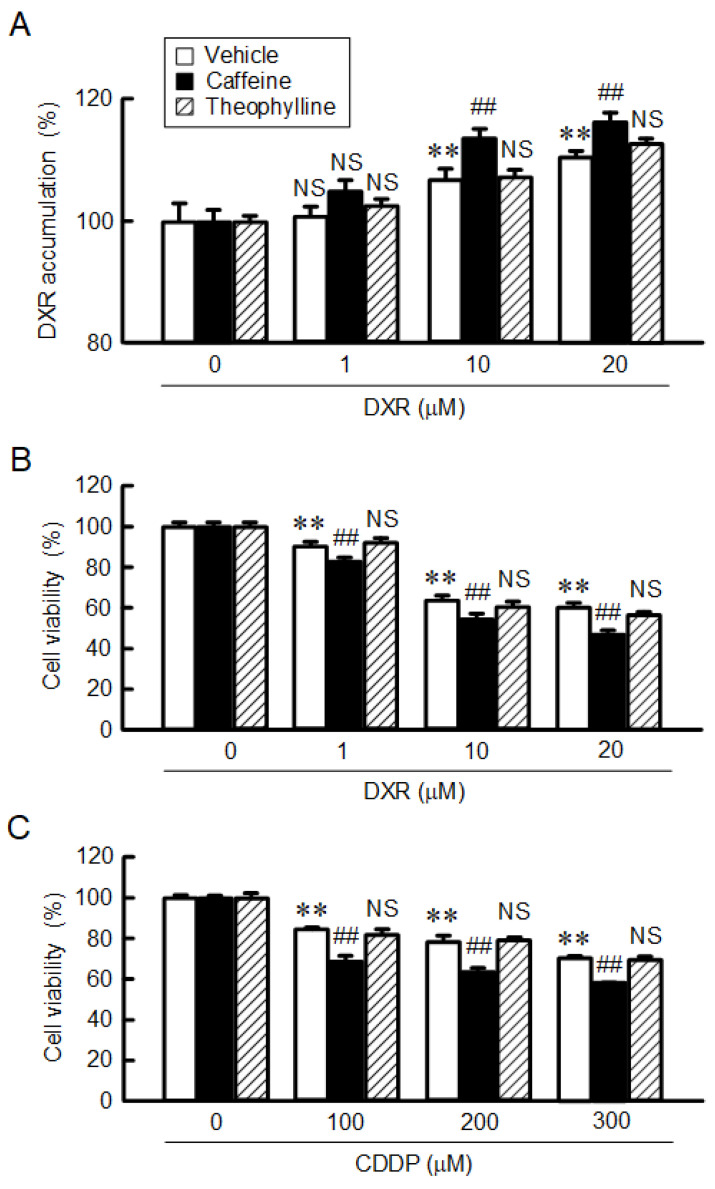
Enhancement of anticancer-drug-induced toxicity in spheroids by caffeine. (**A**) Spheroids were incubated in the absence (vehicle) or presence of 200 μM caffeine or theophylline for 1 h. The fluorescence intensity of DXR in the spheroids was examined using fluorescence microscopy and represented as a percentage relative to 0 μM (autofluorescence). (**B**,**C**) After incubation of the spheroids in the absence (vehicle) or presence of 200 μM caffeine or theophylline 24 h, followed by incubation with DXR and CDDP at the indicated concentrations. Cell viability was measured using a CellTiter-Glo 3D Cell Viability Assay kit and represented as a percentage relative to 0 μM DXR. n = 4–6. ** *p* < 0.01 compared with 0 μM DXR or CDDP. ^##^
*p* < 0.01 compared with vehicle. NS *p* > 0.05 compared with 0 μM or vehicle.

**Figure 9 ijms-23-15447-f009:**
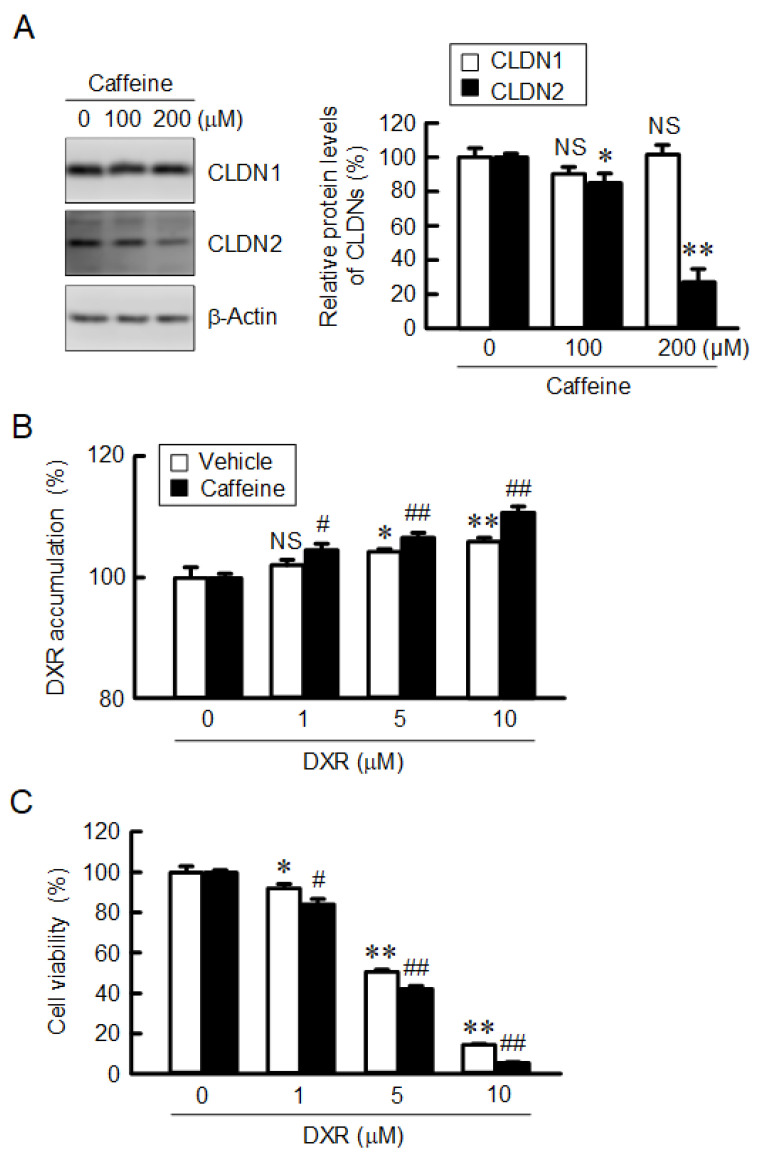
Effect of caffeine on CLDN2 expression and DXR-induced toxicity in RERF-LC-MS cells. (**A**) Cells were incubated with caffeine at the indicated concentrations for 24 h. The expression levels of CLDN1, CLDN2, and β-actin were examined using Western blotting. The protein levels of CLDN1 and CLDN2 are represented as a percentage relative to 0 μM. (**B**) The spheroids were incubated in the absence (vehicle) or presence of 200 μM caffeine for 1 h. The fluorescence intensity of DXR is represented as a percentage relative to 0 μM. (**C**) After incubation of the spheroids in the absence (vehicle) or presence of 200 μM caffeine for 24 h, followed by incubation with DXR at the indicated concentration. Cell viability is represented as a percentage relative to 0 μM DXR. n = 3–8. ** *p* < 0.01, * *p* < 0.05 and NS *p* > 0.05 compared with 0 μM caffeine or DXR. ^##^
*p* < 0.01 and ^#^
*p* < 0.05 compared with vehicle.

## Data Availability

Not applicable.

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
