# Peer review of "Elevation of Anticancer Drug Toxicity by Caffeine in Spheroid Model of Human Lung Adenocarcinoma A549 Cells Mediated by Reduction in Claudin-2 and Nrf2 Expression"

_ijms, 2022, doi:10.3390/ijms232415447_

Round 1
Reviewer 1 Report
In this manuscript, Eguchi and colleagues investigated the effects of several key ingredients of coffee on A549 cells and found that caffeine and theobromine can reduce claudin-2 expression in A549 cells. The finding is interesting, but several issues are raised for authors to clarify before publishing.
1. Since the intake of any compounds over a safety range could lead to side effects, toxicity, and even lethality, could authors review and summarize the side effects and corresponding dose range of each testing compounds? The corresponding drinking volume of coffee to each dose range should also be indicated. Lethal doses of caffeine have been reported at blood concentrations of 80 to 100 ug/ml, which can be reached with ingestion of approximately 10 grams or greater. Personally, I feel 200uM of caffeine (40ug/mL) in plasma is too high and could have some strong adverse effects on humans, especially on the heart. If this is so, warming information should be included in the discussion section.
2. As mentioned by the authors in the discussion, the phosphorylation at serine 208 could determine the stability of claudin-2. The PKA activity and phosphorylation of claudin-2 Serine 208 should be measured and included in Figure 2 or Figure 5.
3. The authors suggested that caffeine and theobromine downregulate claudin-2 expression by promoting lysosomal degradation. Could authors show the colocalization of lysosome and claudin-2 using the confocal microscope and show the results in Figure 5?
4. Despite the downregulation of claudin-2 sensitizing A549 to doxorubicin and cisplatin, is there any effect of the downregulation of claudin-2 on the cell migration and invesion ability of A549 cells? Could authors make some measurements?
5. Including other lung cancer cell lines in the study would be necessary.
Author Response
We thank you very much for your careful reading of our manuscript and valuable comments.
Comment 1
Since the intake of any compounds over a safety range could lead to side effects, toxicity, and even lethality, could authors review and summarize the side effects and corresponding dose range of each testing compounds? The corresponding drinking volume of coffee to each dose range should also be indicated. Lethal doses of caffeine have been reported at blood concentrations of 80 to 100 ug/ml, which can be reached with ingestion of approximately 10 grams or greater. Personally, I feel 200uM of caffeine (40ug/mL) in plasma is too high and could have some strong adverse effects on humans, especially on the heart. If this is so, warming information should be included in the discussion section.
Answer
Following your suggestion, we added the warming information about caffeine in the Discussion. Please see line 314.
Comment 2
As mentioned by the authors in the discussion, the phosphorylation at serine 208 could determine the stability of claudin-2. The PKA activity and phosphorylation of claudin-2 Serine 208 should be measured and included in Figure 2 or Figure 5.
Answer
Following your suggestion, we performed additional experiments. The phosphoserine level of claudin-2 was decreased by caffeine. Please see new figure 5.
Comment 3
The authors suggested that caffeine and theobromine downregulate claudin-2 expression by promoting lysosomal degradation. Could authors show the colocalization of lysosome and claudin-2 using the confocal microscope and show the results in Figure 5?
Answer
Following your suggestion, we performed additional experiments. Claudin-2 was colocalized with Lamp-1, a lysosome marker, in the cells treated with caffeine and chloroquine. Please see new figure 5.
Comment 4
Despite the downregulation of claudin-2 sensitizing A549 to doxorubicin and cisplatin, is there any effect of the downregulation of claudin-2 on the cell migration and invesion ability of A549 cells? Could authors make some measurements?
Answer
Following your suggestion, we performed additional experiments. Cell proliferation and migration were inhibited by caffeine. Please see new figure 6.
Comment 5
Including other lung cancer cell lines in the study would be necessary.
Answer
As shown in figure 9, caffeine induced the reduction of claudin-2 expression and enhanced the doxorubicin-induced toxicity in human lung adenocarcinoma-derived RERF-LC-MS cells.
Reviewer 2 Report
Manuscript ID: ijms-2028276
In the manuscript entitled “Elevation of anticancer drug toxicity by caffeine in spheroid model of human lung adenocarcinoma A549 cells mediated by reduction of claudin-2 and Nrf2 expression”, Eguchi et al. examine caffeine effects on 2D and 3D A549 cell models highlighting reduction of claudin-2 and Nrf2 expression as well as elevation of anti-cancer drug toxicity. Although both experimental design and results are straightforward, the study is merely descriptive without any analysis of which caffeine molecular target is involved in mediating the observed effects. In the discussion, the authors just mention two possibilities, which take into account caffeine mediated PDE inhibition or protein phosphatase activation.
Several molecular targets of caffeine are known since long time such as adenosine receptors, PDEs and ryanodine receptors and new potential targets are continuously discovered. Caffeine is an antagonist of all adenosine receptor subtypes (Ki values: ~10 µM) as well as theobromine albeit with lower potency and theophylline. Theophylline and caffeine are well-known PDE inhibitors (caffeine pIC50 < 4) (Faudone et al. The medicinal chemistry of caffeine. J. Med. Chem. 2021, 64, 7156−7178). The A2B adenosine receptor has been found highly expressed in A549 cells where it is involved in crucial cancer events (Giacomelli et al. The A2B Adenosine Receptor Modulates the Epithelial–Mesenchymal Transition through the Balance of cAMP/PKA and MAPK/ERK Pathway Activation in Human Epithelial Lung Cells. Front. Pharmacol. 2018, 9:54; Kitabatake et al. Involvement of adenosine A2B receptor in radiation-induced translocation of epidermal growth factor receptor and DNA damage response leading to radioresistance in human lung cancer cells. BBA - General Subjects 2020, 1864:129457). On the other hand, the A2B adenosine receptor subtype is also considered an important target for cancer therapy (reviewed in: Gao and Jacobson. A2B Adenosine Receptor and Cancer. Int. J. Mol. Sci. 2019, 20, 5139). To verify whether caffeine elicits its effects by antagonizing adenosine binding to the A2B receptor, a selective antagonist should be used at appropriate concentrations. Indeed, a decrease of A2B receptor activation and signaling could explain both reduced mRNA levels and claudin 2 phosphorylation at Ser208 with enhanced degradation. In fact, this adenosine receptor subtype, which is coupled to Gs and Gq proteins, activates multiple signaling pathways leading to PKA, PKC, and ERK1/2 activation and Ca2+ release from intracellular stores. The use of selective inhibitors of these kinases could also help to decipher the mainly involved pathway. Nevertheless, it is feasible that caffeine can exert its activity by modulating additional targets.
Major comments
1. The introduction and discussion must be improved. The introduction should provide a more complete view of the current knowledge and literature. Sometimes the various topics aren’t well linked together.
2. Figure 7 panel A and B. The difference between vehicle and caffeine spheroid fluorescence does not seem to be so striking as then shown by the bar graph.
3. Figure 7 panel A and B. In panel A spheroids are compact while in panel B spheroids aren’t so. They seem to derive from different cell lines. Is it an effect of the ROS marker, which may reduce cell viability?
Minor comments:
The English language requires some revision.Manuscript ID: ijms-2028276

Author Response
We thank you very much for your careful reading of our manuscript and valuable comments.
Major comments
Comment 1
The introduction and discussion must be improved. The introduction should provide a more complete view of the current knowledge and literature. Sometimes the various topics aren’t well linked together.
Answer
Following your suggestion, we modified the introduction and discussion.
Comment 2
Figure 7 panel A and B. The difference between vehicle and caffeine spheroid fluorescence does not seem to be so striking as then shown by the bar graph.
Answer
Following your suggestion, we replaced the representative images. Please see new figure 7.
Comment 3
Figure 7 panel A and B. In panel A spheroids are compact while in panel B spheroids aren’t so. They seem to derive from different cell lines. Is it an effect of the ROS marker, which may reduce cell viability?
Answer
The representative images were unsuitable. We replaced the other representative images. Please see new figure 7.
Minor comments:
The English language requires some revision.
Answer
Following your suggestion, our manuscript was checked by Medical English Service. We attached the certificate of proofreading.
Round 2
Reviewer 1 Report
All concerns raised for the earlier version of the manuscript have been improved. One thing should be noted the fatalities from caffeine intoxication have been reported with plasma caffeine concentrations of < 40 mg/L which is approximately equal to the dose, 200uM, showing the biological effect in the study. For this reason, the newly added warming information in the discussion section should present the relationship between the study dose and the lethal dose to avoid tragedies.
Author Response
Comment
All concerns raised for the earlier version of the manuscript have been improved. One thing should be noted the fatalities from caffeine intoxication have been reported with plasma caffeine concentrations of < 40 mg/L which is approximately equal to the dose, 200uM, showing the biological effect in the study. For this reason, the newly added warming information in the discussion section should present the relationship between the study dose and the lethal dose to avoid tragedies.
Answer
We thank you very much for your careful reading of our manuscript and valuable comments. Following your suggestion, we modified the Discussion again. Please see line 317.